# Anti-Arthritis Effect through the Anti-Inflammatory Effect of *Sargassum muticum* Extract in Collagen-Induced Arthritic (CIA) Mice

**DOI:** 10.3390/molecules24020276

**Published:** 2019-01-13

**Authors:** Hyelin Jeon, Weon-Jong Yoon, Young-Min Ham, Seon-A Yoon, Se Chan Kang

**Affiliations:** 1Research Institute, Genencell Co. Ltd., Yongin 16950, Korea; iljhl@hanmail.net; 2Biodiversity Research Institute, Jeju Technopark, Namwon 63608, Jeju, Korea; yyjkl@jejutp.or.kr (W.-J.Y.); hijel@jejutp.or.kr (Y.-M.H.); yoonsa33@jejutp.or.kr (S.-AY.); 3Department of Oriental Medicine Biotechnology, Kyung Hee University, Yongin 17104, Korea

**Keywords:** rheumatoid arthritis, collagen-induced arthritis, anti-inflammation, *Sargassum muticum*, functional food

## Abstract

(1) **Background**: Rheumatoid arthritis is a chronic autoimmune disease that causes progressive articular damage and functional loss. It is characterized by synovial inflammation that leads to progressive cartilage destruction. For this reason, research on functional foods that reduce the inflammatory response are under progress. (2) **Methods**: We focused on the anti-inflammatory effects of *Sargassum muticum*, and confirmed the effect of the extract on the collagen-induced arthritis (CIA) DBA/1J mice model. (3) **Results**: The extract was given at concentrations of 50, 100, and 200 mg/kg, and the arthritis score and edema volume of the experimental group were significantly different from the CIA group. The level of interleukin (IL)-6, tumor necrosis factor (TNF)-α, and interferon (IFN)-γ were determined in serum and lymphocytes. The expression of these cytokines in the serum remarkably decreased from *S. muticum* extract (SME)100 mg/kg, and decreased from SME 200 mg/kg in lymphocytes. Also, immunohistochemical analysis of IL-6 and TNF-α in the joints revealed that the inflammatory response was noticeably lower when treated with *S. muticum* extract. (4) **Conclusions**: This study provides results of the experiment of *S. muticum* extract treatment in a mouse model. The treatment was found to contribute to the alleviation of edema and symptoms by reducing the expression of inflammatory cytokines. It was concluded that it may be a useful substance to help in the mitigation of arthritis symptoms.

## 1. Introduction

Rheumatoid arthritis (RA) is a chronic autoimmune disease that causes progressive articular damage, functional loss, and comorbidity. It is characterized by synovial inflammation and hyperplasia, which leads to progressive cartilage and bone destruction [1]. The pathophysiology of RA remains mostly unexplained, but inflammatory mediators such as tumor necrosis factor (TNF)-α, interleukin (IL)-6, and cyclooxygenase (COX)-2 are known to play a pivotal role in the inflammation of synovial membranes and the bone destruction observed in RA [2]. IL-6 is a multifunctional cytokine that usually provides an SOS signal to trigger the host defense by sending out inflammatory signals from localized lesions to the whole body via the bloodstream. Immune deviation as well as local and systemic inflammation associated with RA are mostly explained by the biological activity of IL-6 [3,4]. Microbial infection, tissue damage, and the activation of coagulation cascades are known to induce IL-6 expression [5]. However, the precise mechanism of the initiation and continuous expression of IL-6 in patients with RA is still under research. The crucial role of IL-6 in RA was confirmed by clinical evaluation of the effect of to cilizumab (atlizumab) on RA patients [6]. Therefore, functional food agents or substances that can mediate the downward regulation of these inflammatory components may have a potential for the treatment of RA.

Marine algae or seaweeds contain significant amounts of vitamins, minerals, dietary fibers, proteins, polysaccharides, and various functional polyphenols [7,8]. These nutrient contents are known to vary with species [9], season, temperature [10], and geographical location [11]. Seaweeds, one of the most extensively used functional foods and medicinal herbs, has a long history. It is traditionally consumed as food in East Asian countries, especially in Korea, Japan, China, and Indonesia [12]. It contains a large amount of phycocolloids such as agar, alginate, and carrageenan, which have been utilized as gelling agents and emulsifiers in the food and pharmaceutical industries [13,14]. For this reason, there is a growing interest in finding a pharmacological substance from natural marine plants, and the basic study on the physiological activity of seaweeds, along with a movement to extract these ingredients and develop them into food additives, medicines, and functional foods. Macro-algae can be divided into three types: red, green, and brown, all of which have positive attributes, including protein and small-molecule contents, and can be developed as foods for animals and humans [15]. *Sargassum muticum* (*S. muiticum*) is a large brown algae that belongs to Sargassum and Sargassaceae (Phaeophyceae, Fucales). Apparently, the division of the root, stem, and leaves shows a distinct pattern. The *Sargassum* genus is a very large taxonthat includes over 400 species and is widely distributed across the Pacific coast, Indian Ocean, and Australian coasts [16]. *S. muiticum* from Jeju Island, South Korea, used in this study, is a newly named species distinguished from other plants. There are not enough research results regarding it, although research using this algae is in progress. To date, many researchers have discovered the bioactive nature of peptides and other molecules in macro-algae (and their derivatives), and further discussed the potential use of macro-algae as functional food ingredients. They can be used against cancer [17], oxidative stress [18], inflammation [19], allergy [20], diabetes [21], thrombosis [22], obesity [23], hyperlipidemia [24], hypertensive [25], and other degenerative diseases.

In this study, *S. muticum* markedly inhibited not only inflammation but also clinically evident collagen-induced arthritis. These findings suggest that *S. muticum* can potentially be applied for the clinical treatment of RA.

## 2. Results

### 2.1. Sargassum muticum Suppresses the Clinical Symptoms of Collagen-Induced Arthritis (CIA)

Collagen-induced arthritis (CIA) is an experimental method to derive RA from animal models, and researchers have improved their understanding of RA through this experimental method. We used bovine type II collagen to induce RA in 6-week-old male DBA/1J mice, which was a modification of the method put forth by Azuma et al. [26]. Daily oral administration of *S. muticum* extract (SME) from day 28 to day 98 was found to suppress the incidence and severity of CIA in a dose-dependent manner. Figure 1 shows a representative arthritic clinical score. As can be seen in Figure 1, there was no significant difference found in the arthritis score between the five groups until day 35. There was a significant difference in the arthritis score between the vehicle control (CIA induced group; CIA group) and other groups on the day 42. SME-treated groups showed decreased levels of severity as well. Edema of the feet also increased gradually over time after the induction of arthritis (Figure 2). A decrease in the volume of edema in the SME-treated group compared to the CIA group was witnessed. In addition, mice administered with SME did not exhibit any weight loss (data not shown).

### 2.2. Sargassum muticum Extract Inhibits IL-6, TNF-α, and Interferon (IFN)-γ Production in CIA Mouse Serum

In the CIA mice model that causes RA, inflammation is inevitable and the expression of inflammatory cytokines such as IL-6, TNF-a, and IFN-γ is enhanced [27]. According to the results of this study, the production of inflammatory cytokines increased by more than 3 times to 4.5 times in the CIA group than in the normal group (Figure 3). On the other hand, the expression levels of IL-6 (Figure 3A), TNF-α (Figure 3B), and IFN-γ (Figure 3C) in serum were significantly lower in the experimental groups fed with SME than in the CIA group, depending on the dose amount.

### 2.3. Sargassum muticum Extract Inhibits Splenomegaly and Suppresses IL-6, TNF-α, and IFN-γ Production in CIA Mouse Lymphocytes

If inflammation develops in the body, the burden of the spleen, responsible for the immune response, may increase, resulting in spleen enlargement (splenomegaly) [28]. After the end of the experiment, as a result of measuring the relative weight change of the spleen isolated from each individual, the spleen weight increased in the arthritis-induced groups compared to the normal control group. The spleen weights of the SME groups and positive control group were lower than those of the CIA groups, but there was no significant difference between all of the groups (Figure 4). The expression of cytokines in the splenocytes of CIA DBA/1J mice was similar to that of the production of IL-6, TNF-α, and IFN-γ in serum. According to Figure 5, the CIA group expressed cytokines 2 times to 4.5 times higher than the normal control group. In the case of IL-6 and TNF-α, a significant reduction in expression was also observed at SME 50 mg/kg, but an IFN-γ showed a significant effect at SME 200 mg/kg. SME also decreased the expression of cytokines in lymphocytes—the higher the concentration of SME, the lower the cytokine expression.

### 2.4. Sargassum muticum Extract Suppresses Joint Degradation in CIA Mice

We used the ankle joints of animals treated with the normal control, as well as the CIA group, the SME 200 mg/kg group, and the positive control group (Joins 10 mg/kg) to identify histological changes in the joints of CIA DBA/1J mice. The most common symptoms of RA are the destruction of joints and bones due to inflammation and hyperplasia [29]. As can be seen from the results (Figure 6B), in the normal control group, a smooth cartilage surface was observed and damage such as slit formation was not observed. On the other hand, in the CIA control group, fibrous tissue proliferation was so severe that identifying the exact location of the joint was almost impossible. However, when treated with 200 mg/kg of SME (Figure 6C), it was confirmed that although the surface of the cartilage was irregular due to cartilage damage, the joints were maintained and slit formation was not observed. In the case of the positive control (Figure 6D), the shape of the joint could be seen more clearly compared to the CIA group, but severe cartilage damage and fibrosis were identified. Therefore, we confirmed that the treatment of SME can inhibit the cartilage damage and fibrosis that may occur in RA.

### 2.5. Sargassum muticum Extract Decreases the Expression of Inflammatory Cytokines in CIA Mice

When RA occurs, an inflammatory reaction naturally occurs not only in the blood but also in the damaged joint [30]. Therefore, in this study, we aimed to confirm the expression of IL-6 and TNF-α, which are typical inflammatory cytokines in joints. Immunohistochemical (IHC) analysis was performed using ankle joint tissue fixed with paraffin in order to identify histologic changes. Normal control group experiments were excluded. We used the ankle joints of the CIA group, the SME 200 mg/kg group, and the Joins 10 mg/kg group. In the tissues of the CIA group, the remarkable expression of IL-6 (Figure 7) and TNF-α (Figure 8) were found not only in joints but also in bones. As a result of IHC, deep brown-violet-colored IL-6 and TNF-α accumulated in the articular cartilage, and it was confirmed that the CIA group (panel A) and the positive control group (panel C) were strongly colored. It was also confirmed that the articular cartilage and bone surface had irregular and uneven surfaces due to RA induction. On the other hand, as can be seen from panel B, the shape of the surface of the knee joints of the experimental animals that were treated with SME was relatively maintained and regular. Of course, the accumulation of cytokines was also inhibited compared to panels A and C.

### 2.6. Isolation and Composition Experiments of Sargassum muticum Extract

We analyzed the extracts to determine if any of the components of *S. muticum* extract inhibits the production and accumulation of inflammatory cytokines in the blood, lymphocytes, and joints, in addition to inhibiting joint damage. As can be seen in Figure 9, we carried out the separation experiment using the extract. Sequential fractions of *S. muticum* extract in 70% ethanol were fractionated with *n*-hexane, methylene chloride (CH_2_Cl_2_), ethyl acetate (EtOAc), and *n*-butanol (BuOH). This is a sequential extraction method using the polarity of the solvent. First, 7.0 g of the CH_2_Cl_2_ fraction was adsorbed on Celite and then washed with hexane to obtain H fractions. The H fractions were subjected to silica gel column chromatography (3 cm × 50 cm, Hex:EtOAc:MeOH; 3:1:0.1) to obtain 14 sub-fractions. Nitric oxide (NO) activity-guided fractionation and isolation were executed using the *n*-hexane sub-fraction, and H-10 was the best in inhibiting the NO activity (data not shown). Therefore, we analyzed the components present in H-10. The 10th fraction (H-10) was purified using Prep-thin layer chromatography (TLC) (20 cm × 20 cm, Hex:EtOAc:MeOH; 3:1:0.1), and 10 mg of compound **1** (Apo-9′-fucoxanthinone) was isolated. The structure of compound **1** is shown in the bottom part of Figure 9. The NMR results are as follows.

Compound **1**—Apo-9′-fucoxanthinone: ^1^H-NMR (500 mHz, CDCl_3_) δH (ppm): 5.86 (1H, s, H-8), 5.37 (1H, tt, *J* = 11.4, 4.3 Hz, H-3), 2.33 (ddd, *J* = 2.2, 4.4, 13.1 Hz, H-4a), 2.18 (3H, s, H-10), 2.04 (3H, s, H-15), 2.03 (1H, m, H-2b), 1.52 (dd, *J* = 12.9, 11.3 Hz, H-4b), 1.44 (1H, m, H-2a), 1.43 (6H, s, H-11 and 13), 1.16 (3H, s, H-12); ^13^C-NMR (125 MHz, CDCl_3_) δc (ppm): 209.27 (C-7), 197.87 (C-9), 170.25 (C-14), 118.33 (C-6), 100.83 (C-8), 72.02 (C-5), 67.44 (C-3), 45.07 (C-4), 45.02 (C-2), 36.08 (C-1), 31.68 (C-12), 30.87 (C-13), 28.99 (C-11), 26.49 (C-10), 21.42 (C-15); ESIMS (positive) *m*/*z* 289.19 [M + Na]^+^.

We conducted the following experiments assuming that compound **1** (Apo-9′-fucoxanthinone; Apo-9′), which had been isolated, is effective for anti-inflammation. First, the cytotoxicity of Apo-9′was confirmed. As shown in the Figure 10A, the cytotoxicity of Apo-9′ against Raw264.7 cells was significant at 250 μg/mL. Therefore, we decided to proceed with the experiment at concentrations below 250 μg/mL. After confirming the cytotoxicity of Apo-9′, we used lipopolysaccharide (LPS) to induce nitric oxide (NO) production and prostaglandin E2 (PGE_2_) expression, and to confirm the anti-inflammatory effect of Apo-9′. In addition, cell viability at this time was also confirmed. As can be seen from the blue solid line graph with dots in Figure 10B, slight toxicity appeared at SME 100 μg/mL co-treatment with LPS 1 μg/mL, but there was no significant difference. Apo-9′ was pre-treated for 2 h and then treated with LPS at 1 μg/mL to induce the inflammatory response. As a result, as shown in the bar graph of Figure 10B, the expression of nitric oxide (NO), which is anti-inflammatory mediator, was greatly increased by LPS, but the expression of NO was decreased by Apo-9′ in a dose-dependent manner. The expression level of PGE_2_, another inflammatory mediator, was also identified by ELISA analysis. As can be seen from the results shown in Figure 10C, the low concentration of Apo-9′ significantly inhibited inflammation.

## 3. Discussion

The worldwide prevalence of RA is estimated to be around 0.5–1% with some variation across regions [31]. RA is a chronic systemic autoimmune disease involving the joints and other organs such as the heart and lungs. It is associated with pain, disability, and mortality. It has a heavy impact on people of working age, and it is a major cause of ultimate worklessness [32]. In addition, the disease brings severe long-term economic consequences, including direct costs of medical care, indirect costs of work disability, and interference with social roles, as well as the intangible costs of pain, fatigue, helplessness, loss of self-efficacy, and other psychological difficulties [33]. RA medications can be divided into four categories: nonsteroidal anti-inflammatory drugs (NSAIDs), steroids, disease-modifying anti-rheumatic drugs (DMARDs), and biologic agents. The type of medication that a doctor recommends depends on the severity of the condition and the duration of RA [34]. Because these drugs can cause a variety of serious side effects, it is necessary to develop new materials that are safe and have reduced side effects. For this reason, we sought to find natural materials that are significantly less toxic and adverse, when ingested for the long-term, and are highly effective in treating inflammation.

*Sargassum*, the seaweed under study, is a genus of brown algae (Phaeophyceae), Fucales, Sargassaceae family. It is widely distributed mainly in the temperate zone of the Pacific coast, Indian Ocean, and Australian coasts. It is a very large group that comprises approximately 400 species. So far, anti-inflammatory studies using the *Sargassum* genus have been conducted by many researchers. In particular, *Sargassum muticum* is a species that was newly named in Korea in 2005, distinguishing it from other plants in the family [35]. Through a variety of studies, *S. muticum* has become well known for its effect against cancer [17], oxidative stress [18], inflammation [19], and allergies [20]. In addition, researchers have found that not only *S. muticum* but also other algae extracts of *Sargassum* have antioxidant, anti-diabetic, anti-inflammatory, and anticancer efficacies [18,36,37,38,39,40]. The seaweeds of the *Sargassum* genus, which present various phenolic compounds, all exhibit physiological activities such as anticancer, anti-inflammation, and anti-oxidation activities. In particular, our sample, *S. muticum* extract in 70% ethanol (SME) showed significant anti-rheumatoid arthritis efficacy in CIA mice when crude extracts were provided. The results of this study are as follows. The severity of arthritis was evaluated by the arthritis score, and the severity was alleviated in the SME treatment group. Through histopathological examination (Hematoxylin and eosin staining; H&E staining), the CIA group showed severe joint surface damage, while the SME treatment group showed much less joint destruction and deformation. To determine whether inflammation-related cytokines were altered, IL-6, TNF-α, and IFN-γ levels were measured in serum and lymphocytes, and IHC staining was performed on ankle joints. The cytokines produced by arthritis decreased after SME treatment, and it was confirmed that the high concentration of SME incited a similar response compared to the positive control group.

In conclusion, under the present test conditions, it was found that oral administration of 200 mg/kg or more SME to CIA experimental animals has a positive effect on RA, such as decreased production inflammatory factors and edema. These results were expected due to the presence of the component Apo-9′ in SME, as shown in Figure 9 and Figure 10, which resulted in the anti-inflammatory effect by suppressing pro-inflammatory cytokine expression via NO and PGE_2_. On the other hand, according to Chae et al., Apo-9′ isolated from the extract of *S. muticum* has a potent inhibitory effect on pro-inflammatory cytokine production [41].They confirmed that *S. muticum* extract and Apo-9′ inhibited the production of pro-inflammatory cytokines by attenuating TLR9-dependent AP-1 activation. Of course, additional mechanism studies should be performed to confirm whether the anti-inflammatory effect is exerted through the same mechanism as that reported in the study of Chae et al. Apart from that, however, this study has shown that *S. muticum* extract in 70% ethanol has anti-inflammatory and especially anti-rheumatoid arthritis efficacy in in vitro and in vivo studies. Therefore, we propose the possibility of developing *S. muticum* extract in 70% ethanol into a product for the alleviation of RA symptoms.

## 4. Materials and Methods

### 4.1. Chemicals

RPMI-1640 medium along with fetal bovine serum (FBS) and antibiotics were obtained from GIBCO/Thermo Fisher Scientific Inc. (Waltham, MA, USA). 3-(4,5-dimethylthiazol-2-yl)-2,5-diphenyltetrazolim bromide (MTT) was purchased from Sigma-Aldrich (St. Louis, MO, USA). ELISA assay kit, a product of Bioo Scientific Corporation (Austin, TX, USA), was used to measure the release amount of cytokines. All other chemicals were purchased from Sigma-Aldrich.

### 4.2. Preparation of Sargassum muticum Extract (SME)

*Sargassum muticum* seaweed was collected from a Biodiversity Research Institute, Jeju Technopark located at Jeju Island, and was identified by Weon-Jong Yoon, Jeju Technopark (Jeju, Korea). The voucher specimen (NMR-KR-17-019) was deposited in the Laboratory of Natural Medicine Resources in BioMedical Research Institute, Kyung Hee University. The samples were washed three times with distilled water to remove salts, plants, and sand, and then they were dried without direct sunlight and pulverized. The crude extract was obtained by extracting 2 kg of dried seaweed three times with 70% ethanol for 24 h at room temperature. The extracts were concentrated for 16 h at reduced pressure and 40 °C using a rotary evaporator, and then the crude extract was lyophilized to obtain a powder. This was stored at −20 °C before its use (yield was 3.27%, 65.4 g). When administered orally to experimental animals, the material was mixed with 0.9% normal saline.

### 4.3. Fractionation and Isolation of SME

As can be seen in Figure 9, the extracts were analyzed in the following manner. The extract was dispersed in 1 L of distilled water (H_2_O) and then divided into *n*-hexane fraction, methylene chloride (CH_2_Cl_2_) fraction, ethyl acetate (EtOAc) fraction, *n*-butanol (BuOH) fraction, and a water fraction. The CH_2_Cl_2_ fraction (7.0 g) was adsorbed on Celite and then fractionated by washing it with hexane (hexane/CH_2_Cl_2_ 1:0, 10:1, 5:1, 2:1, 0:1, EtOAc, and methanol). A total of 14 sub-fractions (H-1 to H-14) were obtained by performing silica gel column chromatography (3 cm × 50 cm, hexane/EtOAc/methanol 3:1:0.1). The 10th fraction (H-10) was fractionated by Prep-TLC × 20 cm, hexane/EtOAc 1:1) to separate 10 mg of compound **1** (Apo-9′-fucoxanthinone).

### 4.4. In Vivo Study

#### 4.4.1. Induction of Arthritis

The male DBA/1J mice used in this study are widely used in RA studies. This test system was chosen because of the accumulated abundant test data and their ease of analysis and evaluation [26]. Fifty male DBA/1J mice at 5 weeks of age were purchased from Korea Laboratory Animal Co. (Daejeon, Korea) and were allowed to acclimatize for one week prior to the experiments. The animals were maintained under standard laboratory conditions: a temperature of 21 ± 2 °C, relative humidity of 50 ± 5%, and a normal photoperiod (12 h dark/light). The experimental procedures and animal care protocols were approved by the Animal Care and Use Committee of Kyung Hee University and complied with the Guide for the Care and Use of Laboratory Animals of the National Institutes of Health (NIH Publication No. 85–23).We slightly modified the methods of Mun et al., Banda et al., and Piróg et al. to derive RA [42,43,44]. The group with CIA was designated as the CIA group (vehicle control group). Mice were treated with *S. muticum* extract in 70% ethanol (SME) at different concentrations (50, 100, and 200 mg/kg) to examine the anti-arthritis effects of the extract. Joins was administered at 10 mg/kg as a positive control. Ten animals were assigned to each group. Animal experiments were performed in accordance with the current ethical regulations for animal care and use at Kyung Hee University (KHUAGC-17-005). After one week of acclimatization, bovine type II collagen was dissolved in 10 mM acetic acid at the concentration of 4 mg/kg, and 4 mg/mL Complete Freund’s adjuvant (CFA, Sigma-Aldrich) was mixed 1:1 at 1,000 rpm using a homogenizer. For the induction of arthritis, 100 μg of collagen and CFA mixture were injected intradermally to the root of the mouse tail, and this was boosted again after two weeks to induce arthritis.

#### 4.4.2. Treatment of Extractions

After the induction of arthritis by the injection of collagen, DBA/1J mice were administered orally with SME (50, 100, and 200 mg/kg) mixed with 0.9% normal saline daily. For the therapeutic approach, DBA/1J vehicle control (CIA) mice received 0.9% normal saline alone in both experiments, and Joins (SK Chemicals, Seongnam-si, Kyeonggi-do, Korea), the positive control, was used at the concentration of 10 mg/kg (each group, *n*=10).

#### 4.4.3. Clinical Observation and Evaluation of Arthritis

The occurrence and degree of arthritis were evaluated by checking the presence of edema and redness in four legs of 10 mice. A modified version of the method of Ito et al. [45] was used and after the first injection, and observation was performed once a week for three weeks. After three weeks, severity was assessed and scored by observing two to three times a week, and a maximum of 12 points was given to each individual. The criteria of the score are as follows: Normal, 0; slight swelling and/or erythema, 1; pronounced edematous swelling, 2; ankyloses (state in which the joint is not bent), 3.

#### 4.4.4. Measurement of Paw Swelling

The paw volume was measured using a plethysmometer (IITC Life Science, Woodland Hills, CA, USA) immediately before the induction of arthritis (CIA) and at a certain time after the induction of arthritis.

#### 4.4.5. Isolation and Culture of Lymphocytes from Spleen

After the completion of the experiments, the mouse spleen was separated, weighed, homogenized, and centrifuged at 500× *g* for 20 min to obtain a lymphocyte layer. Red blood cells were eliminated by the addition of a lysis buffer (0.16 mol/L ammonium chloride tris buffer, pH 7.2) at 37 °C for 3 min. Then the primary cells were washed twice with RPMI-1640 culture medium. Cell concentrations were counted and adjusted to 1 × 10^7^ cells/mL. Cell viability was >95% as determined by trypan blue exclusion. Then cells were placed in a 24-well plate and cultured in a 5% CO_2_ incubator for three days.

#### 4.4.6. Measurement of IL-6, TNF-α, and IFN-γ Production in Serum and Splenocytes

After the completion of experiments, blood was collected from the orbit of the mice, and the serum was separated by centrifugation. The prepared lymphocytes were used. The determination of cytokines levels was conducted using MaxDiscovery Mouse ELISA Test Kits from Bioo Scientific according to the manufacturer’s instructions. The determination of IL-6, TNF-α, and IFN-γ levels by a Molecular Devices microplate reader (Menlo Park, CA, USA) was used to measure absorbance at 450 nm.

#### 4.4.7. Histological Staining

Paws were fixed at 10% normal buffered formalin, decalcified in Calci-clear rapid (National Diagnostics, GA, USA), dehydrated, cleared, and embedded in paraffin. Sections with a thickness of 3 μm were cut from the paraffin-embedded tissue blocks and stained with hematoxylin and eosin (H&E) for histological analysis. Histopathological evaluation was carried out by an investigator blinded to the treatment group. The characterization of inflammatory infiltrates was based on morphological criteria. Representative sections are shown at the magnification of ×100.

#### 4.4.8. Immunohistochemical Staining

IHC staining experiments were performed with reference to Lee et al. [46]. The paraffin-embedded tissues were cut into 3-μm thick slides for immunohistochemistry, and the paraffins were removed with xylene. The tissue slides were dipped in phosphate-buffered saline (PBS) and then the endogenous peroxidase was removed using hydrogen peroxide. Primary antibodies were diluted 1:100 with anti-mouse IL-6 and anti-goat TNF-α and incubated overnight at 4 °C. After being washed with PBS, anti-mouse and anti-goat HRP secondary antibodies were reacted, and a mixture of DAB plus hydrogen peroxide was used for color development. Control staining was performed with hematoxylin and mounted with permount and then read by optical microscope. Representative sections are shown at the magnification of ×200.

### 4.5. In Vitro Study

#### 4.5.1. Cell Culture

Raw264.7 murine macrophage cell line was obtained from the American Type Culture Collection (ATCC TIB-71; Rockville, MD, USA). The cells were cultured in Dulbecco’s Modified Eagle’s Medium (DMEM; GIBCO, Grand Island, NY, USA) supplemented with 2 mM L-glutamine, 100 IU/mL penicillin, 100 μg/mL streptomycin, and 10% heat-inactivated fetal bovine serum (FBS). The cells were grown at 37 °C in fully humidified air with 5% CO_2_ and sub-cultured twice per week.

#### 4.5.2. Cytotoxicity Assay

To evaluate the cytotoxicity of samples, 3-(4,5-Dimethylthiazol-2-yl)-2,5-diphenyltetrazolium bromide (MTT) assay was performed according to the previous description, with only a slight modification [47]. Raw264.7 cells (5 × 10^3^ cells/well) were placed in 96-well plates. The medium was replaced with an experimental medium (DMEM with 10% FBS) after one night. To confirm the toxicity of Apo-9′, Raw264.7 cells were treated to 1000 μg/mL and incubated for 24 h. On the other hand, the method for measuring cytotoxicity when inflammation was induced by lipopolysaccharide (LPS) treatment was as follows. Apo-9′ was pre-treated with different doses (0 to 100 μg/mL) of cells for 2 h, and then 1 μg/mL LPS was added. Raw264.7 cells were incubated with SME and LPS for 24 h. MTT reagent was added to each well to the final concentration of 500 μg/mL and incubated for 4 h. The formazan crystals were dissolved in dimethyl sulfoxide (DMSO), and the absorbance at 540 nm was determined with a multireader (TECAN, Infinite 200, Zurich, Switzerland).

#### 4.5.3. Measurement of Nitric Oxide (NO) Production

After 2 h of pretreatment of various fractions of Apo-9′ with different doses (0 to 100 µg/mL), 1 µg/mL LPS-mediated production of NO was measured as nitrite released from Raw264.7 cells at 24 h, as previously described [48]. Briefly, 100 µL of supernatant was combined with an equal volume of Griess reagent (1% sulfanilamide, 0.1% naphthalenediamine dihydrochloride, 2.5% phosphoric acid) and incubated at room temperature for 10 min. The absorbance at 540 nm was determined with an E MAX precise microplate reader (Molecular Devices, Eugene, OR, USA), and nitrite concentrations were calculated from a nitrite standard curve.

#### 4.5.4. Measurement of PGE_2_Production

Raw264.7 cells (1.0×10^6^ cells/mL) cultured in a 96-well plate were pretreated for 2 h with the Apo-9′ component of SME, then stimulated with LPS (1 μg/mL) for 24 h. The PGE_2_ levels were determined using an ELISA kit (R&D Systems) according to the manufacturer’s instructions. The absorbance of each well was measured at 450 nm [49].

### 4.6. Statistical Analysis

All experimental results are expressed as means ± standard error of the mean (SEM) of several independent experiments. Statistical significance of group differences was determined by one-way analysis of variance (ANOVA), and individual differences between the means of groups were analyzed using the student’s *t*-test. The edema rate and inhibition rate are generally expressed as percentages. The statistical method employed was the SPSS 12.1K program, which is a widely used statistical package. *p* values less than 0.05 were regarded as significant; * *p* < 0.05 and ** *p* < 0.01 were considered significant.

## Figures and Tables

**Figure 1 molecules-24-00276-f001:**
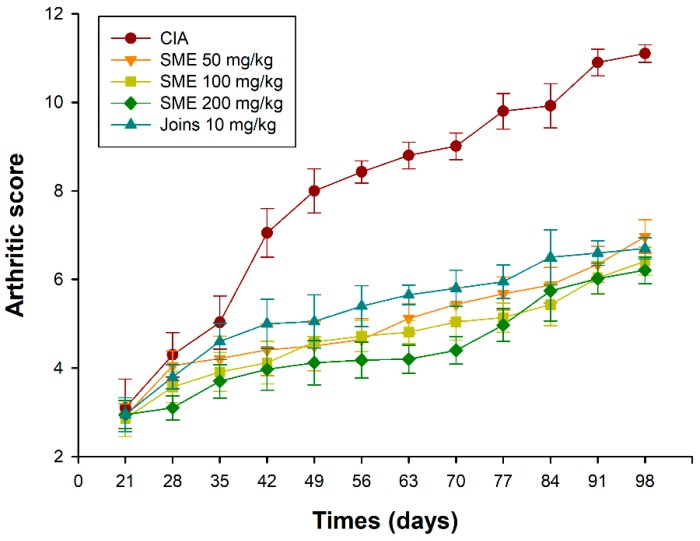
Effects of the *Sargassum muticum* extract on the arthritis score. CIA, collagen-induced arthritis vehicle control group; SME 50 mg/kg, group provided with 50 mg/kg of *S. muticum* extract; SME 100 mg/kg, group provided with 100 mg/kg of *S. muticum* extract; SME 200 mg/kg, group provided with 200 mg/kg of *S. muticum* extract; Joins 10 mg/kg, positive control.

**Figure 2 molecules-24-00276-f002:**
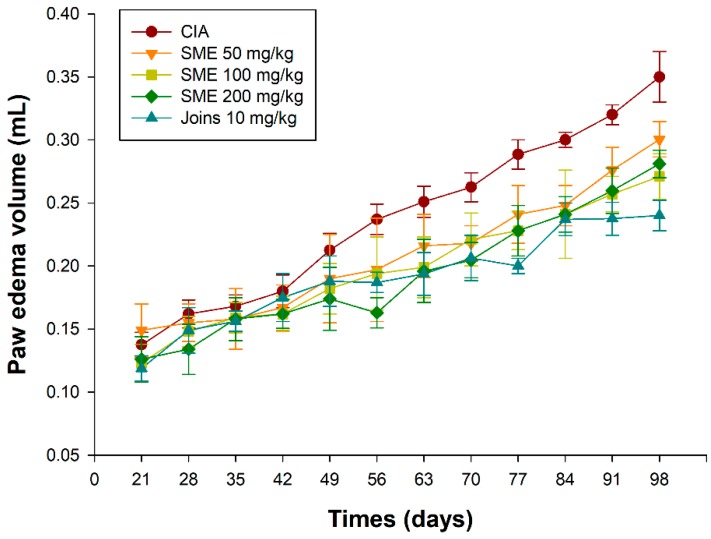
Effects of the *Sargassum muticum* extract on paw edema. CIA, collagen-induced arthritis vehicle control group; SME 50 mg/kg, group provided with 50 mg/kg of *S. muticum* extract; SME 100 mg/kg, group provided with 100 mg/kg of *S. muticum* extract; SME 200 mg/kg, group provided with 200 mg/kg of *S. muticum* extract; Joins 10 mg/kg, positive control.

**Figure 3 molecules-24-00276-f003:**
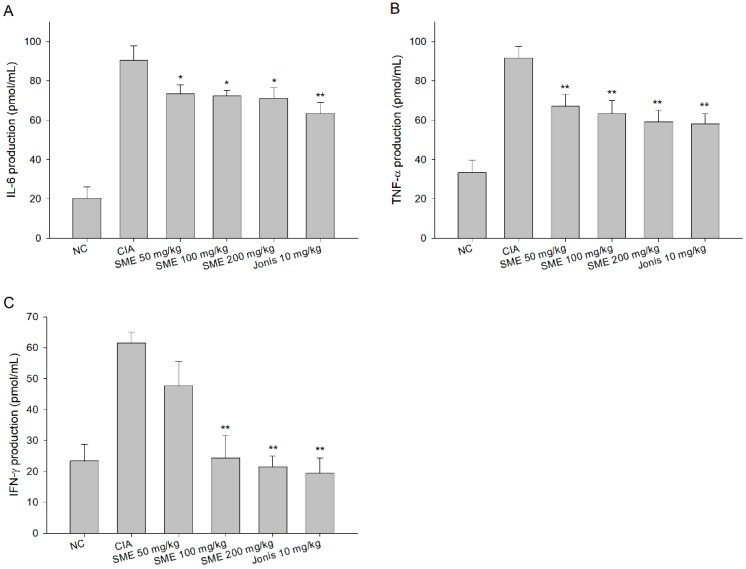
Effects of the *Sargassum muticum* extract on (**A**) interleukin (IL)-6, (**B**) tumor necrosis factor (TNF)-α, and (**C**) interferon (IFN)-γ production in serum. *: significantly different from the CIA group (*p* < 0.05), **: significantly different from the CIA group (*p* < 0.01).

**Figure 4 molecules-24-00276-f004:**
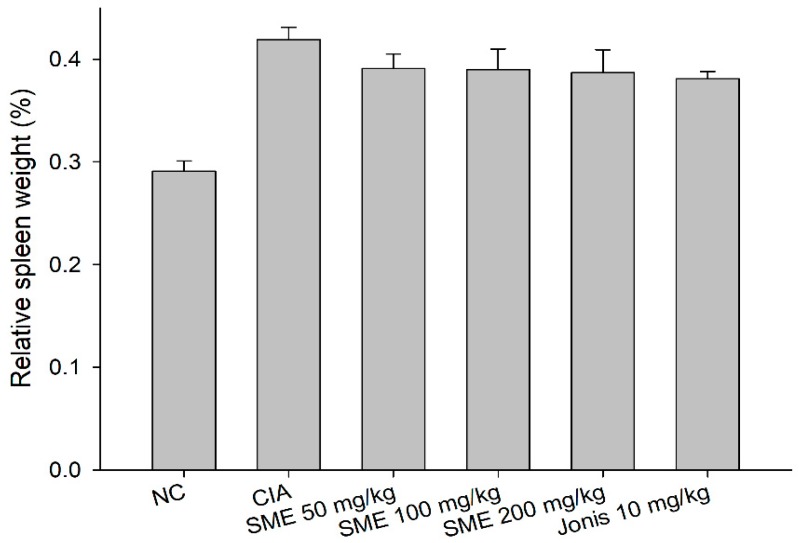
Effects of the *Sargassum muticum* extract on the relative weight of the spleen. There was no significant difference observed.

**Figure 5 molecules-24-00276-f005:**
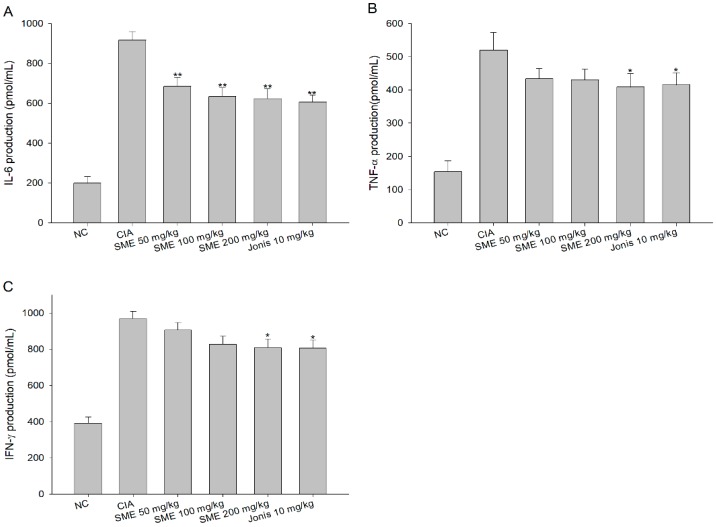
Effects of the *Sargassum muticum* extract on (**A**) IL-6, (**B**) TNF-α, and (**C**) IFN-γ production of lymphocytes. *: significantly different from the CIA group (*p* < 0.05), **: significantly different from the CIA group (*p* < 0.01).

**Figure 6 molecules-24-00276-f006:**
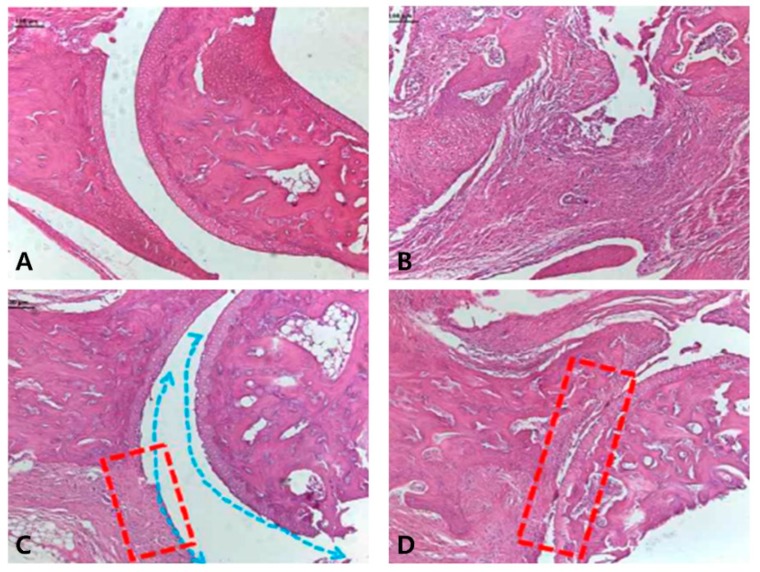
Histological analysis of knee joint. (**A**) Normal control; (**B**) CIA group; (**C**) 200 mg/kg of *S. muticum* extract; (**D**) positive control (Joins, 10 mg/kg). Irregularity of the cartilage surface (blue arrow) was found in (C) and damages to the cartilage (the replacement of fibrotic tissue, red box) were found in (C) and (D). Images are magnified ×100.

**Figure 7 molecules-24-00276-f007:**
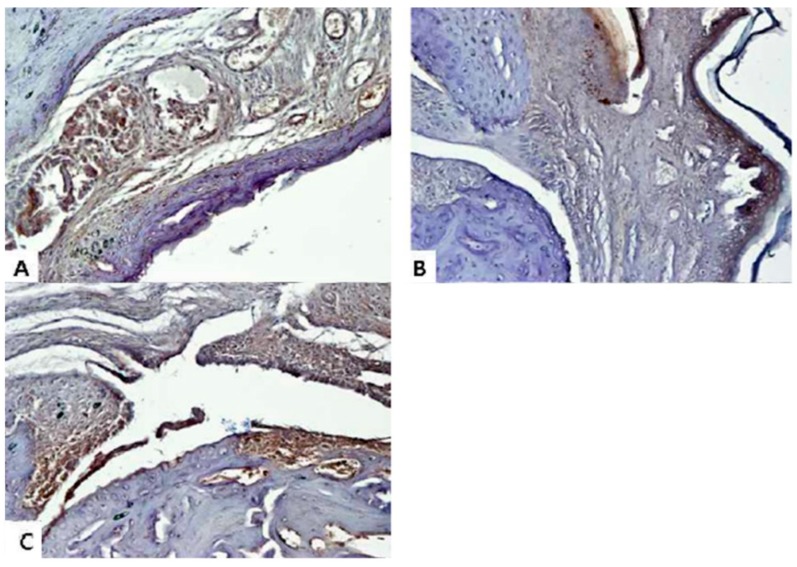
Immunohistochemical analysis of IL-6 from the CIA model knee joint. (**A**) CIA group; (**B**) 200 mg/kg of *S. muticum* extract; (**C**) positive control (Joins, 10 mg/kg). Images are magnified ×200.

**Figure 8 molecules-24-00276-f008:**
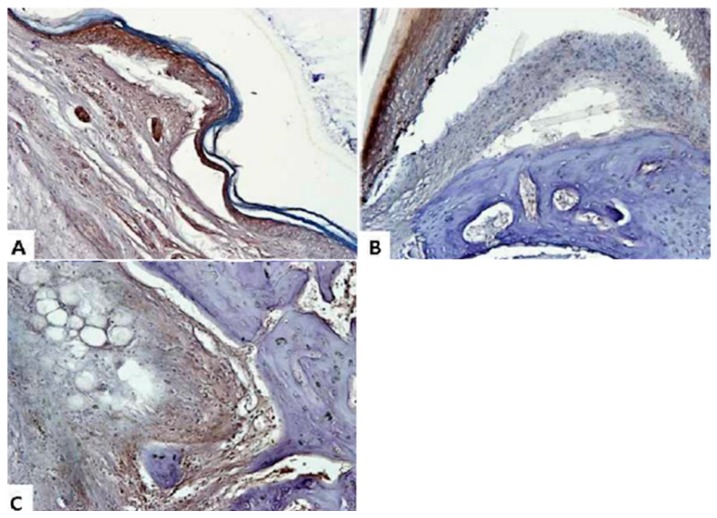
Immunohistochemical analysis of TNF-α from the CIA model knee joint. (**A**) CIA group; (**B**) 200 mg/kg of *S. muticum* extract; (**C**) positive control (Joins, 10 mg/kg). Images are magnified ×200.

**Figure 9 molecules-24-00276-f009:**
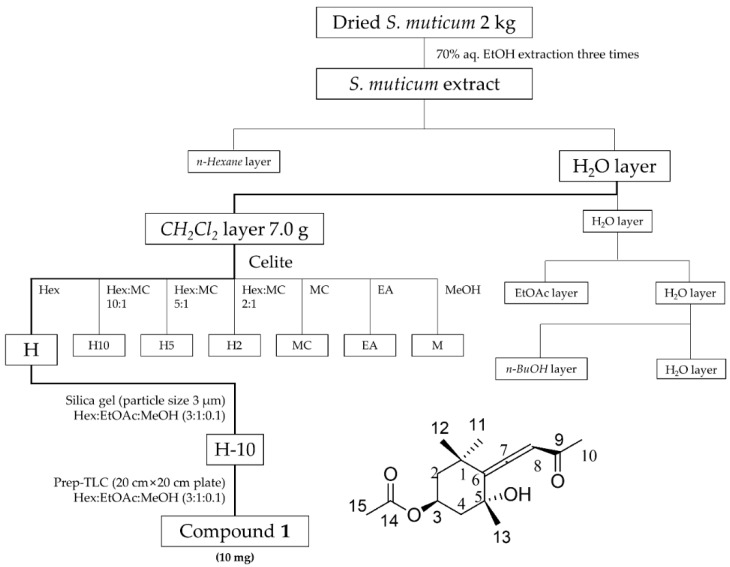
Apo-9′-fucoxanthinone isolated from *S. muticum* extract.

**Figure 10 molecules-24-00276-f010:**
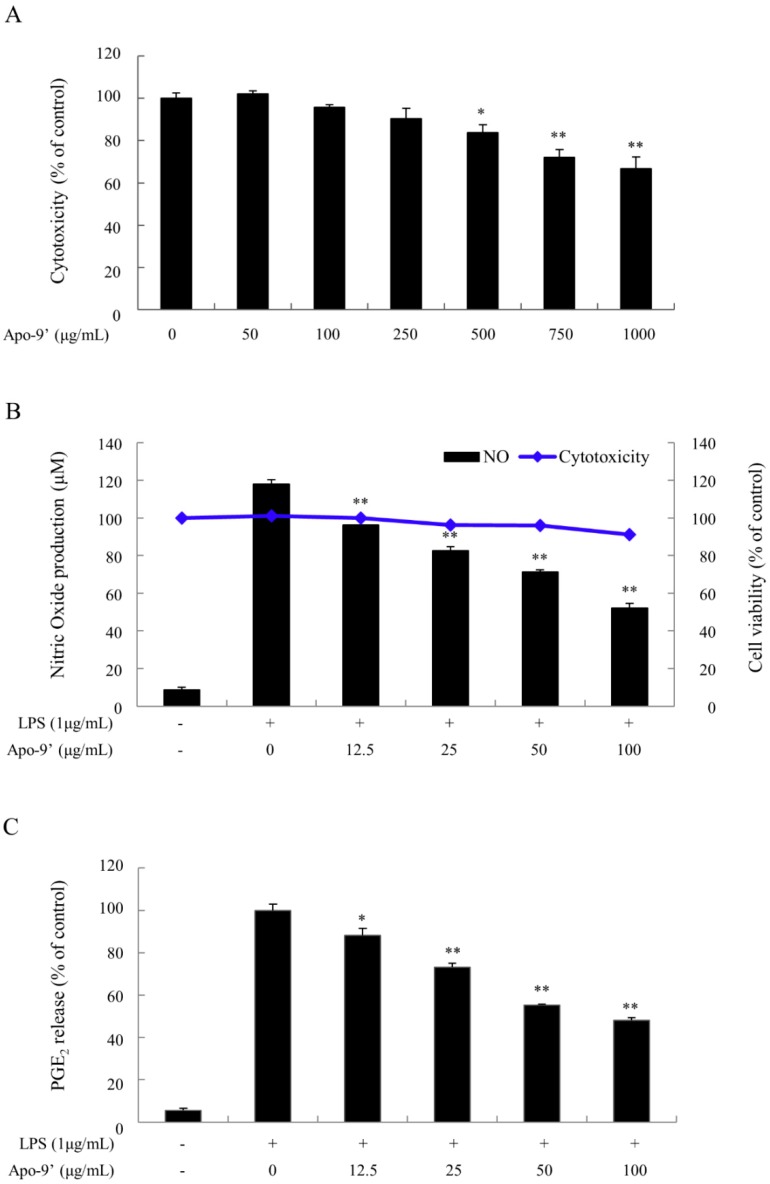
Anti-inflammatory experiments of Apo-9′-fucoxanthinone isolated from *S. muticum* extract. (**A**) Cytotoxicity of Apo-9′-fucoxanthinone in Raw264.7 cells; (**B**) Cytotoxicity and nitric oxide (NO) expression analysis; (**C**) Prostaglandin E2 (PGE_2_) expression analysis. *: significantly different from the CIA group (*p* < 0.05), **: significantly different from the CIA group (*p* < 0.01).

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
