# Peer review of "Anti-Arthritis Effect through the Anti-Inflammatory Effect of Sargassum muticum Extract in Collagen-Induced Arthritic (CIA) Mice"

_molecules, 2019, doi:10.3390/molecules24020276_

Round 1

Reviewer 1 Report

The authors probed the anti-inflammatory effects of Sargassum muticum extract in vivo and in vitro. The extract demonstrated decreases in arthritis score and edema volume in DBA/1J mice, as well as a modest decrease in the expression of cytokines, particularly IFN-g, in serum and in lymphocytes. In addition, immunohistochemical analysis from CIA model knee joints supported these findings. A major isolate (Apo-9’-fucoxanthinone) from the extract demonstrated a dose-dependent decrease in nitric oxide and PGE2 production, although its toxicity was not thoroughly evaluated and appears to be cause for concern.

Overall, this is a well-crafted study and a largely well written manuscript. In terms of the experimentation, my only concern relates to the potential toxicity of the key isolate, Apo-9’-fucoxanthinone. The authors only evaluated its cytotoxicity up to a concentration of 100 mg/mL. In order for this to be a safe and effective food or isolate for anti-inflammatory treatment or arthritis prophylaxis, the TI would have to be at least >10, and preferably much higher. That appears not to be the case based on the increasing toxicity shown in the cell viability assay at doses above 50 mg/mL compared to the IC50 values for NO production and PGE2 release (LD1<IC50, Figure 9B and 9C). I believe the manuscript should be published, but it would be prudent for the authors to comment on this potential toxicity liability.

In addition to that key point, a few minor corrections require attention:

The sentence on lines 22-23 in the Abstract is unclear and should be rewritten.

The last sentence on p 4 is incorrect. The decrease in cytokine production in lymphocytes does not appear to be dose-dependent, as the values are comparable at all administered doses (Figure 5).

“Ch2Cl2” is incorrect and should be changed to ‘CH2Cl2’ (e.g., lines 170, 171, 247)

“The hexane fraction of the celite fraction” (lines 172 and 251) is unclear. Perhaps that could be reworded to, “the hexane wash of the Celite-adsorbed fraction” or similar. Celite should also be capitalized.

On line 171, the material was ‘adsorbed’ onto the Celite surface, not “absorbed” as noted.

For clarity, the phrase ‘with solvents of increasing polarity’ should be added after “fractionated by elution” in line 172.

I believe HRMS data should be included for the Apo-9’-fucoxanthinone isolate to confirm its identity and to support the NMR data.

Is the line a dotted line in Figure 9B (line 185)? It appears to be a blue solid line in the pdf I am viewing.

Line 210, “has” must be changed to ‘have’.

In line 290, “ammonia” is incorrect and should be changed to ‘ammonium’.

What is the MTT assay (line 326)? The authors claim it is previously described, but I do not see such a description. I believe a reference for the assay should also be added.

In summary, the authors describe a well-planned and keenly executed investigation that revealed meaningful results. A few minor corrections are necessary, and I think likely toxicity considerations should be mentioned, but I believe the work is deserving of publication once those concerns are addressed.

Author Response

1) Overall, this is a well-crafted study and a largely well written manuscript. In terms of the experimentation, my only concern relates to the potential toxicity of the key isolate, Apo-9’-fucoxanthinone. The authors only evaluated its cytotoxicity up to a concentration of 100 mg/mL. In order for this to be a safe and effective food or isolate for anti-inflammatory treatment or arthritis prophylaxis, the TI would have to be at least >10, and preferably much higher. That appears not to be the case based on the increasing toxicity shown in the cell viability assay at doses above 50 mg/mL compared to the IC50 values for NO production and PGE2 release (LD1<IC50, Figure 9B and 9C). I believe the manuscript should be published, but it would be prudent for the authors to comment on this potential toxicity liability.

Response: We accepted your opinion and conducted a cytotoxicity study using only Apo-9'-fucoxanthinone. The results are shown in added Figure 10A.

In addition to that key point, a few minor corrections require attention:

2) The sentence on lines 22-23 in the Abstract is unclear and should be rewritten.

Response: Yes, we corrected the sentence as follows; The expression of these cytokines in the serum has remarkably decreased from SME 100 mg/kg, and decreased from SME 200 mg/kg in lymphocytes.

3) The last sentence on p 4 is incorrect. The decrease in cytokine production in lymphocytes does not appear to be dose-dependent, as the values are comparable at all administered doses (Figure 5).

Response: Yes, we corrected the sentences as follows; SME also decreased the expression of cytokines in lymphocytes, the higher the concentration of SM extract, the lower the cytokines expression.

4) Ch2Cl2” is incorrect and should be changed to ‘CH2Cl2’ (e.g., lines 170, 171, 247)

Response: Sorry, we fixed typos.

5) The hexane fraction of the celite fraction” (lines 172 and 251) is unclear. Perhaps that could be reworded to, “the hexane wash of the Celite-adsorbed fraction” or similar. Celite should also be capitalized.

Response: We corrected the sentence as follows;

172 line to 176 line: 7.0 g of the CH2Cl2 fraction was adsorbed on Celite and then washed with hexane to obtain H fractions.

251 line to 285 line: The CH2Cl2 fraction (7.0 g) was adsorbed on the Celite and then fractionated by washing it with hexane.

6) On line 171, the material was ‘adsorbed’ onto the Celite surface, not “absorbed” as noted.

Response: Sorry, it was a typo. And we corrected it.

7) For clarity, the phrase ‘with solvents of increasing polarity’ should be added after “fractionated by elution” in line 172.

Response: We have modified the entire sentence of 172 lines, as follows: 7.0 g of the CH2Cl2 fraction was adsorbed on Celite and then washed with hexane to obtain H fractions.

8) I believe HRMS data should be included for the Apo-9’-fucoxanthinone isolate to confirm its identity and to support the NMR data.

Response: The reviewer suggested adding high-resolution mass spectrometry (HRMS) data, but we have added the electrospray ionization mass spectrometry (ESIMS) data we have. If necessary the HRMS data, we will ask for the test and add the data.

We corrected the sentence as follows; Compound 1; Apo-9′-fucoxanthinone: 1H NMR (500mHz, CDCl3) δH(ppm): 5.86 (1H, s, H-8), 5.37 (1H, tt, J = 11.4,and 4.3 Hz, H-3), 2.33 (ddd, J = 2.2, 4.4 and 13.1 Hz, H-4a), 2.18 (3H, s, H-10), 2.04 (3H, s, H-15), 2.03 (1H, m, H-2b), 1.52 (dd, J = 12.9 and 11.3 Hz, H-4b), 1.44 (1H, m, H-2a), 1.43 (6H, s, H-11 and 13), 1.16 (3H, s, H-12); 13C-NMR (125MHz, CDCl3) δc(ppm): 209.27 (C-7), 197.87 (C-9), 170.25 (C-14), 118.33 (C-6), 100.83 (C-8), 72.02 (C-5), 67.44 (C-3), 45.07 (C-4), 45.02 (C-2), 36.08 (C-1), 31.68 (C-12), 30.87 (C-13), 28.99 (C-11), 26.49 (C-10), 21.42 (C-15); ESIMS(positive) m/z 289.19[M+Na]+

9) Is the line a dotted line in Figure 9B (line 185)? It appears to be a blue solid line in the pdf I am viewing.

Response: What I meant was a blue solid line graph with dots. We acknowledged that the expression was wrong, and corrected the contents. Also, due to the added data, Figures 9B and 9C were changed to Figure 10B and 10C.

The sentences were modified as follows: After confirming the cytotoxicity of Apo-9', we used lipopolysaccharide (LPS) to induce nitric oxide (NO) production and prostaglandin E2 (PGE2) expression, and to confirm the anti-inflammatory effect of Apo-9'. In addition, cell viability at this time was also confirmed. As can be seen from the blue solid line graph with dots in Figure 10B, slight toxicity appeared at SME 100 μg/mL co-treatment with LPS 1 μg/mL, but there was no significant difference. Apo-9′ was pre-treated for 2 h and then treated with LPS at 1 μg/mL to induce the inflammatory response. As a result, as shown in the bar graph of Figure 10B, the expression of nitric oxide (NO), which is anti-inflammatory mediator, was greatly increased by LPS, but the expression of NO was decreased by Apo-9′ dose-dependent. The expression level of PGE2, another inflammatory mediator, was also identified by ELISA analysis. As can be seen from the results shown in Figure 10C, the low concentration of Apo-9′ significantly inhibited inflammation.

10) Line 210, “has” must be changed to ‘have’.

Response: Sorry, we corrected.

11) In line 290, “ammonia” is incorrect and should be changed to ‘ammonium’.

Response: Sorry, we corrected the typo.

12) What is the MTT assay (line 326)? The authors claim it is previously described, but I do not see such a description. I believe a reference for the assay should also be added.

Response: Since MTT analysis is performed basically in cell experiments, we thought that most of the readers would well know what is a MTT analysis. So we omitted the full name of the MTT reagent. But added the full name of MTT to your advice, and added an additional explanation for this experiment.

By the way, the reviewer said, you did not find a reference to the MTT assay in the reference. It is described in the 'cell viability assay' section (179 line) in reference manuscript. https://doi.org/10.1021/acs.jafc.6b00425.

Reviewer 2 Report

Interesting work, raising an important health problem.

But I have some comments:
First - in the manuscript “Anti-inflammatory effects of apo-9'-fucoxanthinone from the brown alga, Sargassum muticum”.   Eun-Jin Yang, Young Min Ham, Wook Jae Lee, Nam Ho Lee and Chang-Gu Hyun. DARU Journal of Pharmaceutical Sciences 201321: 62 https://doi.org/10.1186/2008-2231-21-62  the results of Apo 9 'influence on the production of NO and the release of PGE2 and viability of 264.7 cell culture have already been published – how can you explain this? (Figure 9 in present manuscript and Figure 2 in above cited).

The second main complaint - there is practically no discussion. In addition to summarizing the results, it should be compared reliably with other publications, especially with the anti-arthritis activity of other marine algae. If this work is so innovative that algae have not been tested in this respect, please compare with terrestrial plants. If your results are so good it will be an argument for spreading the consumption of algae in the world.

Thirdly, the test plant, in addition to norisoprenoid (Apo-9 '), also contains phenolic compounds with antioxidant and anti-cancer properties (induction of apoptosis in cancer cells). Ethanol extraction also includes polyphenols. They did not affect the results obtained. Please refer to this.

Why only Apo-9 ' was analyzed. Where does the conclusion that only this compound has anti-inflammatory properties? Have the remaining fractions been checked for this? On what basis the fraction 10 after chromatography was selected. Why is there no reference to literature in this procedure?
In the results, the procedure of fractionating the extract from materials and methods was repeated (lines 169 - 175) - why? The diagram should also be in the methodology.
These issues should be clearly explained.

Lines 170, 171, 247 should be CH2Cl2 instead of Ch2Cl2.

Author Response

1) First - in the manuscript “Anti-inflammatory effects of apo-9'-fucoxanthinone from the brown alga, Sargassum muticum”. Eun-Jin Yang, Young Min Ham, Wook Jae Lee, Nam Ho Lee and Chang-GuHyun. DARU Journal of Pharmaceutical Sciences 201321: 62 https://doi.org/10.1186/2008-2231-21-62 the results of Apo-9' influence on the production of NO and the release of PGE2 and viability of 264.7 cell culture have already been published – how can you explain this? (Figure 9 in present manuscript and Figure 2 in above cited).

Response: In my opinion, the reviewer seems to require clarification because of the fact that the same author is included in the two papers and that the data is similar. Actually, this paper carried out basic research at a time similar to that “Anti-inflammatory effects of apo-9'-fucoxanthinone from the brown alga, Sargassum muticum”. The data looks similar, but I'm sure it's NOT the same data.

However, we have re-tested the inhibition of NO and PGE2 expression using Apo-9'-fucoxanthinone in response to your feedback. Existing results have been replaced by new results (Figure 10B and C).

2) The second main complaint - there is practically no discussion. In addition to summarizing the results, it should be compared reliably with other publications, especially with the anti-arthritis activity of other marine algae. If this work is so innovative that algae have not been tested in this respect, please compare with terrestrial plants. If your results are so good it will be an argument for spreading the consumption of algae in the world.

Response: Yes, we discussed in more depth and added the contents on discussion chapter.

The following are added:

For this reason, we sought to find natural materials that are significantly less toxic and adverse, even with long-term ingestion, and that is highly effective in treating inflammation.

Sargassum, the seaweed we are trying to study, is a genus of brown algae (Phaeophyceae), Fucales, Sargassaceae family and widely distributed in the Pacific coast, Indian Ocean and Australian coasts, mainly in the temperate zone. It is a very large group comprises approximately 400 species. So far, anti-inflammatory studies using Sargassum genus have been done by many researchers. In particular, Sargassum muticum is a newly named species in Korea in 2005, distinguishing it from other plants in the family [35]. Through a variety of studies, S. muticum is well known that this alga has effective against cancer [17], oxidative stress [18], inflammation [19], and allergy [20]. In addition, researchers have found that not only S. muticum but also other algae extracts of Sargassum genus also have antioxidant, antidiabetic, anti-inflammatory and anti-cancer efficacy [36-41]. The seaweeds of the Sargassum genus, which present the various phenolic compounds are may inevitable that they exhibit physiological activities such as anticancer, anti-inflammation, and anti-oxidation. In particular, our sample, S. muticum 70% ethanol extract showed significant anti-rheumatoid arthritis efficacy in CIA mice when crude extracts were provided. The results of this study are as follows.

In conclusion, under the present test conditions, it has been found that oral administration of 200 mg/kg or more SME to CIA experimental animals has a positive effect on RA, such as decreased production inflammatory factors and edema. These results were expected to be due to the presence of the component Apo-9' in the SME, as shown in Figures 9 and 10 the resulted in the anti-inflammatory effect by suppressing pro-inflammatory cytokine expression via NO and PGE2. On the other hand, according to Chae et al., Apo-9' isolated from the extract of S. muticum have a potent inhibitory effect on pro-inflammatory cytokine production [42]. They confirmed that S. muticum extract and Apo-9' inhibited the production of pro-inflammatory cytokines by attenuating TLR9-dependent AP-1 activation. Of course, additional mechanism studies should be performed to confirm whether the anti-inflammatory effect is exerted through the same mechanism as the study of Chae et al. Apart from that, however, this study has shown that S. muticum 70% ethanol extract has anti-inflammatory, especially anti-rheumatoid arthritis efficacy in vitro and in vivo studies. Therefore, we propose the possibility of developing S. muticum 70% ethanol extract as a product for alleviating RA symptoms.

3) Thirdly, the test plant, in addition to norisoprenoid (Apo-9'), also contains phenolic compounds with antioxidant and anti-cancer properties (induction of apoptosis in cancer cells). Ethanol extraction also includes polyphenols. They did not affect the results obtained. Please refer to this.

Response: As the reviewer said, the brown alga, Sargassum muticum, which we studied, contains a variety of phenolic compounds. In the case of simple crude extract (70% ethanol extract), there are many phenolic compounds in addition to the apo-9'-fucoxanthinone studied by us. Therefore, it is natural that anti-oxidation, anti-inflammation, and anticancer activity by these phenolic compounds will be shown. We did not think it was necessary to enumerate the various activities of the extracts and the components present therein. Our focus is that Apo-9'-fucoxanthinone, which is present in the 70% ethanol extract of S. muticum (SME) has an anti-inflammatory effect, in addition, SME ingestion may help CIA animals alleviate symptoms of arthritis.

4) Why only Apo-9 ' was analyzed. Where does the conclusion that only this compound has anti-inflammatory properties? Have the remaining fractions been checked for this? On what basis the fraction 10 after chromatography was selected. Why is there no reference to literature in this procedure?

Response: I am sorry that the description of this part is not clear. Let me explain the question raised by the reviewer.

In order to separate the physiologically active substances (mainly non-polar substances) dissolved in the water, the components dissolved in the H2O layer were separated into CH2Cl2 and then fractionated into n-hexane. We obtained 14 sub-fractions using n-hexane and proceeded with NO activity guided fractionation and isolation, confirming that the 10th sub-fraction, H-10, had the best inhibitory effect on NO activity. For this reason, we have analyzed the components in H-10. If necessary, I will add this experiment result as supplement data.

The above information has been added to the section on 2.6. Isolation and composition experiments of Sargassum muticum extract.

Contents were modified as follows:

As can be seen in Figure 9, we carried out the separation experiment using the extract. Sequential fractions of S. muticum 70% ethanol extract was fractionated with n-hexane, methylene chloride (CH2Cl2), ethyl acetate (EtOAc), and n-butanol (BuOH). This is a sequential extraction method using the polarity of the solvent. And 7.0 g of the CH2Cl2 fraction was adsorbed on Celite and then washed with hexane to obtain H fractions. The H fractions were subjected to Silica gel column chromatography (3 cm × 50 cm, Hex: EtOAc: MeOH; 3: 1: 0.1) to obtain 14 sub-fractions. NO activity guided fractionation and isolation was carried out using n-hexane sub-fraction, and the best inhibition of NO activity was H-10 (data not shown). Therefore, we analyzed the components present in H-10. The 10th fraction (H-10) was purified using Prep-TLC (20 cm × 20 cm, Hex: EtOAc: MeOH; 3: 1: 0.1), and 10 mg of compound 1 (Apo-9′-fucoxanthinone) was isolated. The structure of compound 1 was the same as the bottom part of Figure 9, also the NMR results were as follows.

5) In the results, the procedure of fractionating the extract from materials and methods was repeated (lines 169 - 175) - why? The diagram should also be in the methodology.

These issues should be clearly explained.

Response: The description in the lines 169-175 was written to explain the picture in Figure 9A. It has been retained as it adds the ‘fractionation and isolation’ associated with this explanation. I also think that the methodological content of the fraction of the extract is well explained in section 4.3.

6) Lines 170, 171, 247 should be CH2Cl2 instead of Ch2Cl2.

Response: Sorry, we fixed typos.

Reviewer 3 Report

Abstract section:

Authors should better describe the nature of the employed extract (aqueous, alcoholic...).

Line 22: correct "has remarkably decreased from" in "was remarkably decreased by".

Abstract and Introduction sections: please describe in extenso the reported acronyms (TNF, IL6 etc...) at their first appearance in the text.

Line 43: As regards the debate about the role of IL-6 expression, authors should also consider that IL-6 could act as an anti-inflammatory cytokine via downregulation of TNFalpha gene expression (Cite: Starkie et al. Exercise and IL-6 infusion inhibit endotoxin-induced TNF-alpha production in humans. FASEB J. 2003 May;17(8):884-6.; Menghini et al. A natural formulation (imoviral™) increases macrophage resistance to LPS-induced oxidative and inflammatory stress in vitro. J Biol Regul Homeost Agents. 2014 Oct-Dec;28(4):775-82.).

At the end of Introduction section, authors should better describe the endpoints of the study.

Experimental section:

Authors should include the approval number of animal experimental procedure.

They should include the total number of randomized animals and the animal number per group, as well.

The analysis reported in 4.5 paragraphs should be  better descibed in details.

Results:

Figures 3 and 5: The ANOVA P value should be included in the legends alongside with the P values related to post hoc test.

Legend of Figure 9: AUthors should include ANOVA and post hoc test P values.

Thoughout the manuscript:

Authors should underline that they administered an alcoholic extract.

Discussion section:

This section is reported in a very limited way. There is a lack of comparison with previous studies.

The improvement of this section is mandatory for manuscript acceptance.

Author Response

1) Authors should better describe the nature of the employed extract (aqueous, alcoholic...).

Response: The extract is stated to be 70% ethanol extract.

2) Line 22: correct "has remarkably decreased from" in "was remarkably decreased by".

Response: We corrected the sentences as follows; The expression of these cytokines in the serum has remarkably decreased from SME 100 mg/kg, and decreased from SME 200 mg/kg in lymphocytes.

3) Abstract and Introduction sections: please describe in extenso the reported acronyms (TNF, IL6 etc...) at their first appearance in the text.

Response: Added a description of the first emerging acronyms.

4) Line 43: As regards the debate about the role of IL-6 expression, authors should also consider that IL-6 could act as an anti-inflammatory cytokine via downregulation of TNFalpha gene expression (Cite: Starkie et al. Exercise and IL-6 infusion inhibit endotoxin-induced TNF-alpha production in humans. FASEB J. 2003 May;17(8):884-6.; Menghini et al. A natural formulation (imoviral™) increases macrophage resistance to LPS-induced oxidative and inflammatory stress in vitro. J Biol Regul Homeost Agents. 2014 Oct-Dec;28(4):775-82.).

Response: However, in normal cases, IL-6 and TNF-a are more likely to act as pro-inflammatory cytokines and this trend has also been observed in our study. I understand what the reviewer wants to say, but that's a bit contrary to our research. In our study, IL-6, TNF-α, and IFN-γ acted as pro-inflammatory cytokines, and SME significantly decreased these cytokines expression in serum and lymphocytes.

At the end of Introduction section, authors should better describe the endpoints of the study.

Experimental section:

5) Authors should include the approval number of animal experimental procedure.

Response: We added the approval number of animal experimental procedure. It was KHUAGC-17-005.

6) They should include the total number of randomized animals and the animal number per group, as well.

Response: Yes, it was added. Note, the 10 animals were assigned to each group. We have purchased 50 male DBA/1J mice for this study.

7) The analysis reported in 4.5 paragraphs should be better descibed in details.

Response: Yes, there was additional content on in vitro assays (especially, cytotoxicity assay).

Results:

8) Figures 3 and 5: The ANOVA P value should be included in the legends alongside with the P values related to post hoc test.

Response: Statistical significance of group differences as determined with one-way analysis of variance (ANOVA), and individual differences between the means of groups were analyzed using the student t-test. I would like to ask if there is a reason why it should not be expressed as p-value of student t-test. I do not think our statistical processing method is wrong.

9) Legend of Figure 9: AUthors should include ANOVA and post hoc test P values.

Response: Similar to the above. I would like to ask if there is a reason why it should not be expressed as p-value of student t-test.

Thoughout the manuscript:

10) Authors should underline that they administered an alcoholic extract.

Response: We have already defined that SME is 70% ethanol extract of S. muticum.

Discussion section:

11) This section is reported in a very limited way. There is a lack of comparison with previous studies. The improvement of this section is mandatory for manuscript acceptance.

Response: Yes, we have added the following to our discussion chapter.

The following are added:

For this reason, we sought to find natural materials that are significantly less toxic and adverse, even with long-term ingestion, and that is highly effective in treating inflammation.

Sargassum, the seaweed we are trying to study, is a genus of brown algae (Phaeophyceae), Fucales, Sargassaceae family and widely distributed in the Pacific coast, Indian Ocean and Australian coasts, mainly in the temperate zone. It is a very large group comprises approximately 400 species. So far, anti-inflammatory studies using Sargassum genus have been done by many researchers. In particular, Sargassum muticum is a newly named species in Korea in 2005, distinguishing it from other plants in the family [35]. Through a variety of studies, S. muticum is well known that this alga has effective against cancer [17], oxidative stress [18], inflammation [19], and allergy [20]. In addition, researchers have found that not only S. muticum but also other algae extracts of Sargassum genus also have antioxidant, antidiabetic, anti-inflammatory and anti-cancer efficacy [36-41]. The seaweeds of the Sargassum genus, which present the various phenolic compounds are may inevitable that they exhibit physiological activities such as anticancer, anti-inflammation, and anti-oxidation. In particular, our sample, S. muticum 70% ethanol extract showed significant anti-rheumatoid arthritis efficacy in CIA mice when crude extracts were provided. The results of this study are as follows.

In conclusion, under the present test conditions, it has been found that oral administration of 200 mg/kg or more SME to CIA experimental animals has a positive effect on RA, such as decreased production inflammatory factors and edema. These results were expected to be due to the presence of the component Apo-9' in the SME, as shown in Figures 9 and 10 the resulted in the anti-inflammatory effect by suppressing pro-inflammatory cytokine expression via NO and PGE2. On the other hand, according to Chae et al., Apo-9' isolated from the extract of S. muticum have a potent inhibitory effect on pro-inflammatory cytokine production [42]. They confirmed that S. muticum extract and Apo-9' inhibited the production of pro-inflammatory cytokines by attenuating TLR9-dependent AP-1 activation. Of course, additional mechanism studies should be performed to confirm whether the anti-inflammatory effect is exerted through the same mechanism as the study of Chae et al. Apart from that, however, this study has shown that S. muticum 70% ethanol extract has anti-inflammatory, especially anti-rheumatoid arthritis efficacy in vitro and in vivo studies. Therefore, we propose the possibility of developing S. muticum 70% ethanol extract as a product for alleviating RA symptoms.

Round 2

Reviewer 2 Report

The authors' replies are sufficient. The text has been corrected and supplemented. Please, check the compatibility of the text edition with the requirements of the magazine very carefully. Statistical signifficance level should be marked under the picture.

Reviewer 3 Report

The manuscript has been significantly improved after revision.